# Replication study of "Data-Driven Methods for Balancing Fairness and Efficiency in Ride-Pooling"

## Reproducibility Summary

**Scope of Reproducibility**

We evaluate the following claims related to fairness-based objective functions presented in [4] : (1) For the four objective functions, the success rate in the worst-off neighborhood increases monotonically with respect to the overall success rate. (2) The proposed objective functions do not lead to a higher income for the lowest-earning drivers, nor a higher total income, compared to a request-maximizing objective function. (3) The driver-side fairness objective can outperform a request-maximizing objective in terms of overall success rate and success rate in the worst-off neighborhood. This means that this objective, whilst reducing the spread of income, also positively impacts rider fairness and profitability.

**Methodology**

The code provided by [4] was used as a base for our re-implementation in PyTorch. We evaluate the claims by the original authors by (a) replicating their experiments, (b) testing for sensitivity to a different value estimator, (c) examining sensitivity to changes in the preprocessing method, and (d) testing for generalizability by applying their method to a different dataset.

**Results**

We reproduced the first claim since we observed the same monotonic increase of the success rate in the worst-off neighborhood with respect to the overall success rate. The second claim we did not reproduce, since we found that the driver-side fairness objective function obtains a higher income for the lowest-earning drivers than the request-maximizing objective function. We reproduced the third claim, since the driver-side objective function performs best in terms of overall success rate and success rate in the worst-off neighborhood, and also reduces the spread of income. Changes of the value estimator, preprocessing method and even dataset all led to consistent results regarding these claims.

**What was easy**

The paper is written engagingly and the theoretical sections, in particular, give a clear description of the problem setup and objectives. The paper is also accompanied by an open-source code base, which supports reproduction efforts.

**What was difficult**

The provided code lacks a script to preprocess raw data, which is required to reproduce the experiment, nor was the preprocessed data openly available. Additionally, complex code structure and scarce commenting complicated replication.

**Communication with original authors**

Due to the absence of preprocessed data, we contacted the authors, who quickly provided the requested data.

# 1  Introduction

Ride-pooling platforms match independent drivers with multiple riders. This matching is performed by machine learning algorithms, which are designed to maximize company profit. The profit motive behind these algorithms can cause unfairness among drivers and riders, for instance by unequally distributing rides between drivers, or by servicing requests originating in some neighborhoods at lower rates [2, 3, 4].

The paper *Data-Driven Methods for Balancing Fairness and Efficiency in Ride-Pooling* [4] (henceforth referred to as "the paper" or "the authors") explores the tradeoff between rider fairness, driver fairness, and total income (i.e. company profitability) by measuring fairness and profitability metrics across simulations using four objective functions. These objective functions are maximized by an algorithm that matches rider requests to drivers, using actual request data from the New York Yellow Taxi dataset. The following metrics are studied: a) the percentage of all requests that are serviced (the *overall success rate*), b) the success rate in the neighborhood with the lowest local success rate (the *success rate in the worst-off neighborhood*), and c) the distribution of income across drivers. The overall success rate is used as a proxy for company profitability, whereas the success rate in the worst-off neighborhood and the income of the least-earning drivers are measures of rider and driver fairness, respectively.

The matching algorithm, introduced by [6], uses a Markov decision process (MDP) in combination with a neural value estimator to match rides to drivers non-myopically, that is, with awareness of future events that could impact the value of a match. The paper's algorithm requires a strongly connected graph (i.e. street network) on which the taxis operate, precomputed routes and travel times between all pairs of nodes (i.e. intersections), and a dataset of requests, each containing an origination node, a destination node, and the time the request was issued.

The paper compares two profitability-focused objective functions and two fairness-focused objective functions. The profitability-focused request objective maximizes the total number of requests serviced during the simulation, given by the sum of ongoing requests $p_i$ and completed requests $s_i$, for each driver $i$:

$$o_{request}(R, W) = \sum_{i=1}^{n}(|p_i| + |s_i|),\tag{1}$$

where $R$ and $W$ are sets respectively containing the states of the drivers, and all previously unaccepted and accepted requests. The profitability-focused income objective maximizes the total income of all drivers:

$$o_{income}(R, W) = \sum_{i=1}^{n}\pi_i,\tag{2}$$

where $\pi_i$ is the income of driver $i$, which is made up of a constant part and a variable part that depends on the distance of the trip. The rider fairness objective maximizes profit whilst minimizing the variance of the success rate across all neighborhoods $j$:

$$o_{rider}(R, W) = -\lambda Var(\frac{h_j}{k_j}) + \sum_{i=1}^{n}\pi_i,\tag{3}$$

where $h_j$ is the number of serviced requests originating in neighborhood $j$, $k_j$ is the total number of requests originating in $j$, and $\lambda$ a hyperparameter moderating the regularization. The driver-side fairness objective maximizes profit whilst minimizing the variance of the income across all drivers $i$:

$$o_{driver}(R, W) = -\lambda Var(\pi_i) + \sum_{i=1}^{n}\pi_i.\tag{4}$$

# 2  Scope of reproducibility

We focus our reproducibility study on the claims related to the fairness-based objective stated in the previous section. These claims (henceforth referred to as *the claims*) can be summarized as follows:

  1. For the four objective functions, the success rate in the worst-off neighborhood increases monotonically with respect to the overall success rate.

2. The proposed objective functions do not lead to a higher income for the lowest-earning drivers, nor a higher total income, compared to a request-maximizing objective function. This extends claim 1, suggesting that a profit motive generally leads to increased fairness for drivers and riders alike.

3. The driver-side fairness objective (4) can outperform a request-maximizing objective (1) in terms of overall success rate and success rate in the worst-off neighborhood. This means that this objective while reducing the spread of income, also positively impacts rider fairness and profitability.

The authors have demonstrated claim 3 to hold only for 50 drivers. For 200 drivers they find the opposite, namely that equation (1) outperforms equation (4) in terms of fairness as well as profitability metrics.

We evaluate these claims by testing whether they still hold under a variety of modifications to the experimental setup. We conduct four experiments that deviate increasingly from the exact setup of the paper: (a) the experiment of the paper is reproduced using the author's preprocessed data, (b) the non-myopic neural value estimator is replaced by a myopic greedy estimator, (c) the experiment of the paper is replicated using data generated with our own preprocessing method, and (d) the experiment is applied to a different dataset (New York Green Taxi dataset), using our own preprocessing method. Note that all experiments except for (b) use the neural value estimators also used in the original paper.

# 3 Methodology

This section explains the methodology used for the four experiments we carried out. An overview of the used code is provided, followed by descriptions of the model, the datasets and the hyperparameters. Finally, we outline the experimental setup and state the computational resources needed to perform the experiments.

## 3.1 Code

The code provided by the authors, which is largely based on the code by [6], was used as a base for our re-implementation in PyTorch. The provided code was sufficient to reproduce the experiments performed in the original paper, after the authors emailed the preprocessed data that they used in the paper. This data consists of the graph of Manhattan, travel times, routes, and rider requests from the New York Yellow Taxi dataset mapped to nodes on the graph. Neither the code to generate this data nor the data itself are publicly available.

To solve the integer linear problem that determines which set of actions is assigned to which driver, the original code uses the callable library CPLEX 12.8. However, at the time of this replication study, the free edition of CPLEX does not suffice to train the required models, so the no-cost academic edition of CPLEX 20.1[1] was used instead.

In addition to an exact reproduction, we examine the paper's claims' robustness against a different method of preprocessing the same raw data. The paper uses the method described in [5] to preprocess the raw trip data. However, this method is computationally demanding and complex to implement. We developed an algorithm that generates routes and travel times on a graph, but that differs in two ways from [5]. First, we use travel time estimates from OpenStreetMap (OSM), corrected by a multiplication constant, equal to the mean ratio between the actual travel time and the OSM estimate of the travel time, computed over all trips in the dataset.

Second, unlike the method used in the paper, we do not compute the Dijkstra algorithm for each pair of nodes, which is an $O(n^2)$ approach, where $n$ is the number of nodes in the graph. For the street network of Manhattan ($n \approx 4000$), this can take days on a typical laptop without GPU acceleration. Instead, our routing algorithm invokes Dijkstra on a total of $\approx 500,000$ pairs of randomly sampled nodes, which yields a coverage of $\approx 60\%$ of all $n^2 = 16M$ routes, because all subroutes of each route are also optimal routes. The optimal routes between the remaining 40% of node pairs are approximated by setting each remaining route $(n, m)$ equal to the concatenation of subroutes $(n, p)$ and $(p, m)$, for a predecessor $p$ selected from a set of predecessors of node $m$, for which routes $(n, p)$ and $(p, m)$ are known and yield the lowest total travel time.

Lastly, to test the generalizability of the methods used in the paper, we performed additional simulations using trips in Brooklyn from the New York Green Taxi dataset. The algorithm described above is used to generate routes and travel times between all nodes, given the graph of Brooklyn's street network.

---

[1]`https://pypi.org/project/cplex/`

## 3.2 Model descriptions

The original paper has adapted the model that incorporates an MDP to assign a set of actions to each vehicle from [6]. An overview of this algorithm is provided in Appendix A. The model makes use of a neural value estimator that assigns a value to each possible set of actions. In the original paper, the objective functions (equations 1, 2, 3, 4) that this model aims to maximize are varied, along with the number of drivers (50 and 200 drivers are used).

The input to the neural value estimator [6] is composed of the current location and path of the vehicle, the permissible delay, the current epoch, the number of other vehicles in the vicinity and the number of requests that were placed in the current epoch. By using an LSTM, this value estimator can take into account non-myopic considerations like the possibility that a future rider request will appear along the route of a current rider request, therefore increasing the value of the current request. The location and path features are embedded using pretrained embeddings, which were computed by a separate network that was trained to predict travel times between any two nodes in the graph.

## 3.3 Datasets

The original experiments are conducted on the New York Taxi dataset [1], available at [2], consisting of pickup and drop-off locations and times for Yellow Taxi passengers from March 23rd to April 1st and from April 4th to April 8th, 2016.

The generalization experiments are conducted on the New York Green Taxi dataset [3], consisting of pickup and drop-off locations and corresponding times for Green Taxi passengers in the months February, March and June. We take the subset of trips in Brooklyn, to keep the size of the graph manageable and similar to the graph of Manhattan used in the original paper. Whereas Yellow taxis in practice only serve the business district of downtown Manhattan, the Green taxis serve all of New York. This makes the Green Taxi dataset a good choice to test the generalization of results concerning rider-side metrics, because of differences in affluence between various districts of New York. In addition, Brooklyn is a different geographical area and the Green Taxi dataset contains a different distribution of requests, compared to the Yellow Taxi dataset.

## 3.4 Hyperparameters

No hyperparameter search was done for this replication study. Instead, the hyperparameters provided by [4] were used to adhere to the original experimental setup as much as possible. This meant that for the driver-side fairness objective function we set $\lambda$ to be $\frac{4}{6}$, and for the rider-side fairness objective function, $\lambda$ was set to be $10^9$. Additionally, we set the constant costs of a ride to \$5, the capacity of a car to 4 riders, the number of neighborhoods to 10, and we batch rider requests per minute.

## 3.5 Experimental setup

For each experiment containing a neural value estimator (section 2), the model is trained for each combination of objective function (equations 1, 2, 3, 4) and number of drivers ($\in \{50, 200\}$). For the specific packages and their versions that were used to obtain our results, we refer to our codebase [4]. Reproducing the experiments by [4] means that we trained all models on 3 days of data, except for the rider-side fairness objective (equation 3) models, which were trained on 2 days of data. All models were evaluated on one day of data.

In order to evaluate the claims, the following metrics are of interest: the overall success rate, the success rate in the worst-off neighborhood (also called *minimum request success rate*), and the income per driver. To compare the objective functions, three plots are generated per experiment, in line with the reporting of the original paper: 1) the minimum request success rate as a function of the overall success rate for the simulation with 50 drivers, 2) the same plot for the simulation with 200 drivers, and 3) the distribution of income across drivers for each objective function for the simulation with 200 drivers.

During the exact reproduction of the paper's experiments, we observed that the neural model is not trained (seemingly inadvertently) in the experiment with 50 drivers. The number of training examples for the neural net grows in the

---

[2] https://www1.nyc.gov/site/tlc/about/tlc-trip-record-data.page

[3] https://www1.nyc.gov/site/tlc/about/tlc-trip-record-data.page

[4] https://anonymous.4open.science/r/rideshare-replication-815C

number of drivers and the number of days of training data. The authors included a minimum threshold of training examples, which is not met by the experiment with 50 drivers. However, the algorithm silently executes, therefore using an untrained randomly-initialized neural network. Importantly, the value assigned to each action is a linear combination of a deterministic term and the output of the (untrained) neural net. Therefore, even though the network's outputs are random in the case of 50 drivers, the computed values are not. This finding motivated the experiment where the neural value estimator is replaced by a greedy value estimator.

To test the claims under a different data preprocessing method, we run an experiment using the same raw data as the original paper, but with our self-developed preprocessing method (i.e. a graph of Manhattan with routes and travel times computed as outlined in section 3.1). Apart from the data preprocessing, this experiment is identical to the experiment in the paper. With this, we aim to investigate the sensitivity of the claims and the method used by [4] to different data preprocessing methods and in particular to a different travel time computation, which is inherently noisy.

Our final experiment uses the Green Taxi dataset and our own preprocessing method in order to study the generalizability of the work by [4]. The models for the Green Taxi dataset were trained on 6 days of data, except for the rider-side fairness objective (equation 3) models, which were trained on 4 days of data. All these models were evaluated on two days of data. For all experiments using our preprocessing method, two new sets of pretrained embeddings were trained for the corresponding graphs of Manhattan and Brooklyn.

## 3.6  Computational requirements

All experiments were run on one Nvidia GeForce 1080Ti GPU. Training and evaluating a model on the Yellow Taxi dataset for 50 drivers took roughly 2.5 hours, whereas training and evaluating one for 200 drivers took 6-7.5 hours. For the rider-fairness objective function models, which were only trained on two days of data, runtimes were two-thirds of this. The experiments that use a greedy non-neural value estimator took an hour less for 200 drivers and roughly the same amount of time for 50 drivers. For the Green Taxi dataset, more days of data were used to train on, since this dataset contains fewer requests per day. Here, the model took roughly 1 hour to train and evaluate for 50 drivers and 10 hours for 200 drivers. Computing embeddings of size 100 for all $\approx 4000$ locations took approximately 10 hours. Finding shortest paths and computing travel times between all pairs of locations took 1.5 hours on a laptop with an Intel i5 processor [5].

# 4  Results

In this section, we present the results of the reproduction of the original paper, the replacement of the neural value estimator by the greedy non-neural value estimator, the replication using our data preprocessing method, and the generalization experiment using the Green Taxi dataset. For each experiment, we evaluate whether the claims listed in Section 2 are supported by the presented results.

## 4.1  Reproduction of the original experiment

The first claim states that objective functions which improve the overall success rate for riders also improve the success rate in the worst-off neighborhood. The figures that the authors use to support this are displayed in Appendix B and our reproduced results in Figures 1a and 1b. If objective functions that improve the overall success rate also improve the success rate in the worst-off neighborhood, we should see the success rate in the worst-off neighborhood monotonically increase as the overall success rate increases. This is indeed a pattern that we see in the figures from the original paper, as well as in our figures.

Second, the authors claim that no objective function raises wages for the lowest-earning drivers or raises the total income, compared to the requests objective function (equation 1). In the original paper, results related to the income distribution used to support this claim are only obtained for 200 drivers, as can be seen in Appendix B. In our reproduction the function that maximizes total income at 200 drivers is also the requests objective function. However, the maximum income for the least-earning drivers is obtained with the driver-side fairness objective function, as displayed in Figure 1c.

---

[5]Specifically, a MacBook Pro (13-inch, 2020) with 1.4 GHz Quad-Core Intel Core i5 CPU

The final claim states that the driver-side fairness objective function (equation 4) outperforms the requests objective function (equation 1) in terms of overall success rate, success rate in the worst-off neighborhood, and reducing the spread of income. These first two results are only obtained for 50 drivers, as seen in the figures in Appendix B, and the last result only for 200 drivers. We reproduce all three of these results. In Figure 1a, we see that both the highest overall success rate and the highest success rate in the worst-off neighborhood are obtained with equation (4). Finally, we find that equation (4) reduces the spread of income compared to (1), as can be seen from Figure 1c.

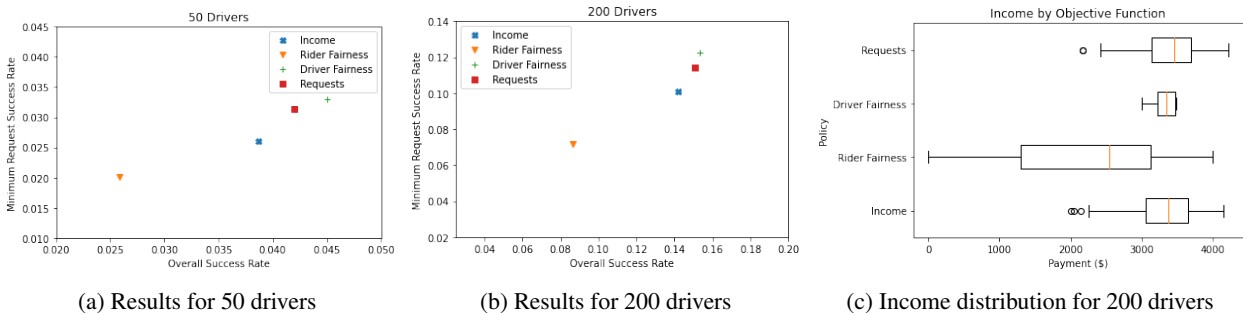

| (a) Results for 50 drivers | (b) Results for 200 drivers | (c) Income distribution for 200 drivers |

Figure 1: Results of the reproduction study. In subplots 1a and 1b, each marker represents an objective function. As in the original paper, success rates are small, because the number of drivers is much smaller than the number of riders. In subplot 1c, the income by objective function is shown.

## 4.2 Results beyond original paper

In addition to reproducing the results in the original paper, we performed supplementary experiments to test the generalizability of the methods used in the original paper. In subsection 4.2.1 we discuss the results of replacing the neural value estimator with a greedy value estimator. Further, in subsection 4.2.2 we examine the results of using our preprocessing method. Lastly, in subsection 4.2.3 we discuss the results obtained using the Green Taxi dataset.

### 4.2.1 Results using a greedy non-neural value estimator

These results are obtained by replacing the neural non-myopic value estimator with a myopic greedy value estimator. This serves both to test the added value of the neural estimator, and to test the sensitivity of the claims to a different estimation method. Interestingly, the results obtained with the greedy estimator (Figure 2) closely resemble the results obtained with the neural estimator (Figure 1), even in the case of 200 drivers where the neural estimator does train (as opposed to the case with 50 drivers). This suggests that, given the limited training examples for 200 drivers and 3 days of taxi data, the neural model provides little added benefit over a myopic greedy estimator. By extension, this experiment reaches the same conclusions: claims 1 and 3 are replicated, but claim 2 is not.

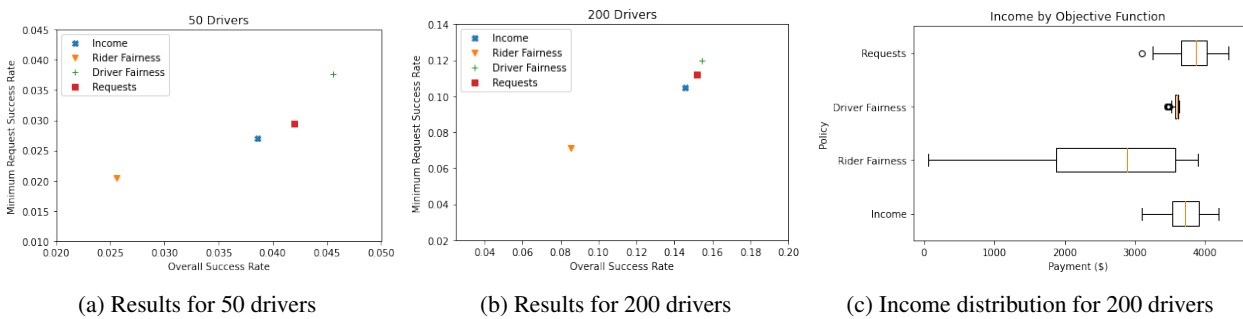

| (a) Results for 50 drivers | (b) Results for 200 drivers | (c) Income distribution for 200 drivers |

Figure 2: Results of using a greedy value estimator. Apart from using a different value estimator, this experiment uses the exact same setup as the original paper (and that we use in our reproduction in section 4.1).

### 4.2.2 Results using our preprocessing method

To examine the sensitivity to changes in the preprocessing method, we compare the results of this experiment to the results of the reproduction. The obtained results, as displayed in Figure 3, are similar to the reproduced results. We see the same monotonic increase in Figures 3a and 3b as we saw in Figure 1 for the reproduction. Similarly, the income distribution (Figure 3c) shows that objective (4) obtains the highest income for the lowest-earning drivers and reduces the spread of income. Therefore, we can conclude that these results support claim 1 and claim 3, but do not support claim 2, and that the method proposed by the authors is robust to changes in the preprocessing method.

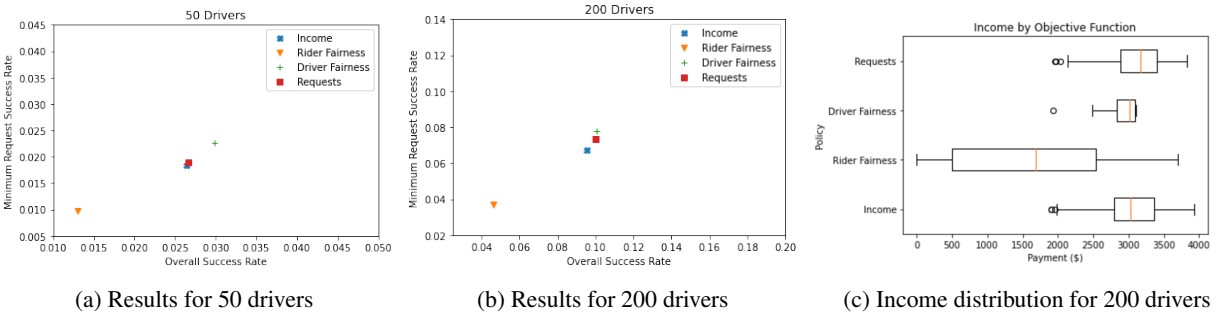

(a) Results for 50 drivers     (b) Results for 200 drivers     (c) Income distribution for 200 drivers

Figure 3: Results of using our preprocessing method. Apart from preprocessing the raw Yellow Taxi data differently, this experiment uses the exact same setup as the original paper (and that we use in our reproduction in section 4.1).

### 4.2.3 Results on Green Taxi dataset

In order to analyze the generalizability of the original paper, we use the results obtained on the Green Taxi dataset to see if the claims described in Section 2 hold for a different dataset. These results (Figure 4) are similar to the ones obtained on the Yellow Taxi dataset in both the reproduction experiment (Figure 1) and the experiment using our preprocessing (Figure 3). This experiment again supports claims 1 and 3, but does not support claim 2.

Important to note is that the success rates are much higher compared to those obtained on the Yellow Taxi dataset. This is because the Green Taxi demand in Brooklyn is significantly less than the Yellow Taxi demand in Manhattan. Hence, with an equal number of taxis a greater share of requests can be met. The fact that the findings of this experiment are consistent with the previous experiments, all of which have much lower success rates, provides confidence that the claims also generalize to realistic success rates (i.e. success rates approaching 100%, as is expected from real taxi companies).

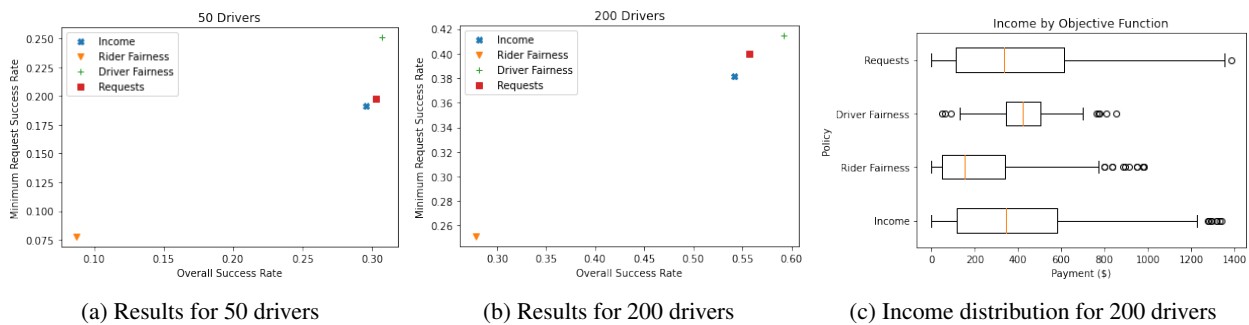

(a) Results for 50 drivers     (b) Results for 200 drivers     (c) Income distribution for 200 drivers

Figure 4: Results for the Green Taxi Dataset. In subplots 4a and 4b, each marker represents an objective function. These results were obtained by averaging the success rates obtained for the two evaluation days. In subplot 4c, the income by objective function is shown. Here, the drivers from each evaluation day are treated separately.

# 5  Discussion

To conclude, our results support the first claim, since we find that the success rate in the worst-off neighborhood increases monotonically with respect to the overall success rate for both 50 and 200 drivers. Unlike the original paper, we do not find that the request-maximizing objective function (equation 1) maximizes income for the lowest-earning drivers, so we do not support the second claim. Instead, we find that at 200 drivers the driver-side fairness objective function (equation 4) obtains the highest income for the lowest-earning drivers.

We support the third claim since our results show that the driver-side fairness objective function yields the greatest overall success rate, the highest success rate in the worst-off neighborhood, and reduces the spread of driver income. Indeed, our results provide even stronger support for this claim than the original paper. Unlike the paper, we have found the driver-side objective to provide the best success rates in both cases of 50 and 200 drivers. Furthermore, this objective lifts the income of the least-earning drivers above what they make under the request-maximizing objective. This makes the driver-fairness objective attractive in all respects, and an interesting subject for further research. One avenue to explore is *why* the driver-fairness objective produces greater success rates than the request-maximizing objective, even though the latter's sole purpose is to maximize the success rate.

The obtained results are generally consistent across the four conducted experiments, showing that the claims which are supported by our results (claims 1 and 3) are robust and relatively insensitive to a range of reasonable changes to the experimental setup. This provides confidence that these claims and the corresponding objective functions generalize well beyond the precise setups in which we and [4] tested them.

One limitation of our work, which is also shared by the original paper, is that the success rates across all experiments are unrealistically low, because the number of drivers (50 or 200) is insufficient to meet demand (there are more than 10,000 Yellow taxis in Manhattan). This creates an abundance of possible actions for each driver that is not representative of the competition that exists for real-world taxi services. Running experiments in a setup where success rates are more realistic would be a worthwhile additional generalization experiment. Such experiments may require more computational resources than some researchers, ourselves included, have access to. However, our results for the Green Taxi dataset already mitigate concerns that the claims would not generalize to greater success rates; the claims were upheld in this setup with success rates of over 50%.

## 5.1  What was easy

The paper is written engagingly and the theoretical sections in particular give a clear description of the problem setup and objectives. The paper is also accompanied by an open-source codebase with their implementation, which is extremely helpful to obtaining accurate reproductions.

## 5.2  What was difficult

Even though the code was sufficient to reproduce the experiments done in the original paper, it was cluttered at times. It contains functions that are never used, as well as print statements solely used to check if a certain point in the code is reached without errors. Additionally, the code used to create the location embeddings creates embeddings of size 10, when embeddings of size 100 are expected by the model. When reimplementing the code in PyTorch, most difficulty was experienced when having to feed masked data into an LSTM backwards, as a result of how masking is implemented in PyTorch. Further, various important details could only be found in the code, such as which days they used to train the model on and the used epoch duration. One of the most important details that was missing is the notion that the model does not train for 50 drivers and three days of Yellow Taxi data, as it will not train without enough examples. Further, the data to perform the original experiments and the script to preprocess the raw data were not publicly available. After contacting the authors, they provided us with the missing data. Despite what is stated in the paper, approximately 1% of the routes in this data were not computed. These omitted routes, however, were not present in any of the rider requests and therefore did not pose a problem to this research.

## 5.3  Communication with original authors

We contacted the authors because the data and the embeddings used to perform the original experiments are not publicly available. The authors provided the missing data quickly and expressed willingness to help with further queries.

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

# A  NeurADP Algorithm

---

**Algorithm 1** NeurADP(N,T)

---
1:  Initialize: replay memory $M$, Neural value function $V$ (with random weights $\theta$)
2:  **for** each episode $1 \leq n \leq N$ **do**
3:      Initialize the state $s_0^n$ by randomly positioning vehicles.
4:      Choose a sample path $\xi^n$
5:      **for** each step $0 \leq t \leq T$ **do**
6:          Compute the feasible action set $F_t$ based on $s_t^n$.
7:          Solve the ILP in Table 1 to get best action $a_t^n$. Add the Gaussian noise for exploration.
8:          Store $(r_t^n, F_t)$ as an experience in $M$.
9:          **if** t % updateFrequency == 0 **then**
10:              Sample a random mini-batch of experiences from $M$
11:              **for** each experience $e$ **do**
12:                  Solve the ILP in Table 1 with the information from experience $e$ to get the objective value $y^e$
13:                  **for** each vehicle $i$ **do**
14:                      Perform a gradient descent step on $(y^{e,i} - V(r_t^{i,n}))^2$ with respect to the network parameters $\theta$
15:                  **end for**
16:              **end for**
17:          **end if**
18:          Update: $s_t^{a,n} = T^a(s_t^n, a_t^n), s_{t+1}^n = T^\xi(s_t^{a,n}, \xi_{t+1}^n)$
19:      **end for**
20: **end for**

---

# B  Results of the original paper

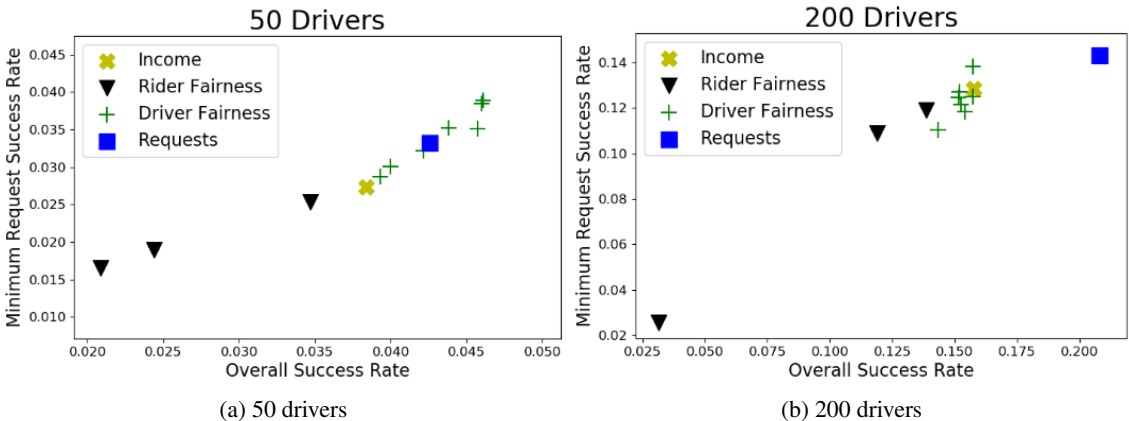

(a) 50 drivers                    (b) 200 drivers

Figure 5: The results of the original paper [4]. The overall success rate is the rate of accepted requests in all neighborhoods, and the minimum request success rate is the minimum rate of accepted requests in a neighborhood. Each data point of the same symbol refers to a different (unspecified) value of $\lambda$.

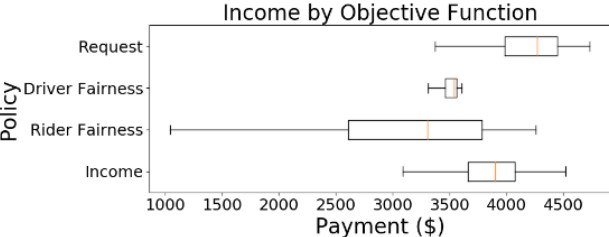

Figure 6: Results of the original paper [4]. The income distribution for 200 drivers.

