# OpenReview forum: "Replication study of "Data-Driven Methods for Balancing Fairness and Efficiency in Ride-Pooling""
_ML_Reproducibility_Challenge/2021/Fall — RC2021_

### Official Review · Reviewer_4wuo · 2022-03-01
**Review: Replication study of "Data-Driven Methods for Balancing Fairness and Efficiency in Ride-Pooling"**

**Rating:** 7
**Confidence:** 4

**Review:**

This study report was well-written, clear, and extends the original paper in interesting ways via generalization experiments. The sampled Djikstra preprocessing approach and extension to Green Taxis are particularly notable.

Questions:
- In Section 3.1, what are the differences in CPLEX versions (free vs. academic/full versions) that make it suitable or unsuitable for this analysis?
- In Section 3.6, the GPU used was described but not the CPU/other computational aspects.

Minor points:
- The package version numbers in the `environment.yml` and `preprocessing_environment.yml` files include hashes after the version numbers that may cause compatibility issues depending on the user's system. It would be better to just use the version numbers.
- The prefix in the `environment.yml` and  `preprocessing_environment.yml` files contains some de-anonymizing information.

---

### Official Review · Reviewer_QGnJ · 2022-03-07
**Thorough and well-written replication/extension**

**Rating:** 9
**Confidence:** 4

**Review:**


The original paper by Raman, Shah, and Dickerson considers the matching problem at the heart of ride-sharing platforms: how can drivers and passengers be matched. They develop cost functions that optimize for forms of fairness and show that they can also improve the service/profitability of the service as a whole.

This is a well-written and nuanced replication study. The authors first reimplement the code used in the original paper, using code published by its author and replicate the original results, then explore four increasingly divergent extensions: different data preprocessing, a different dataset, and a different neural value estimator. The last extension was motivated by the authors’ discovery of a logic error in the original code. I found these extensions interesting and practical, especially since the preprocessing steps used for this data are quite computationally expensive.

 The replication paper is well-written and presents both the results and the motivation for additional experiments clearly. My one minor nit is the author’s use of the term “worst-off” neighborhood; something like “worst-served” would be clearer, even if it does deviate from the phrasing in the original paper.

---

### Meta-Review · Area_Chair_LWii · 2022-04-08

**Recommendation:** Accept
**Confidence:** 4

**Metareview:**

A great reproducibility study where the authors extend the empirical analysis beyond the original work.  The overall presentation is great and with some minor issues which can be fixed.

---

### Decision · Program_Chairs · 2022-04-09

**Decision:**

Accept

**Comment:**

Following the recommendation of reviewers and meta-reviewer, the paper is accepted for ML Reproducibility Challenge 2021, and will be published in the upcoming special edition of ReScience Journal.